# Residual effect of defeated stripe rust resistance genes/QTLs in bread wheat against prevalent pathotypes of *Puccinia striiformis* f. sp. *tritici*

**Harpreet Singh**[1], **Jaspal Kaur**[2]*, **Ritu Bala**[2], **Puja Srivastava**[2], **Achla Sharma**[2], **Gomti Grover**[2], **Guriqbal Singh Dhillon**[3], **Rupinder Pal Singh**[2], **Parveen Chhuneja**[4], **Navtej Singh Bains**[2]

**1** Department of Fruit Science, PAU, Ludhiana, India, **2** Department of Plant Breeding & Genetics, PAU, Ludhiana, India, **3** Department of Biotechnology, Thapar Institute of Engineering and Technology, Patiala, India, **4** School of Agricultural Biotechnology, PAU, Ludhiana, India

* jassu75@pau.edu

**Data Availability Statement:** All relevant data are within the manuscript and its Supporting Information files.

## Abstract

The periodic breakdowns of stripe rust resistance due to emergence of new virulent and more aggressive pathotypes of *Puccinia striiformis* f. sp. *tritici* have resulted in severe epidemics in India. This necessitates the search for new and more durable resistance sources against stripe rust. The three bread wheat cultivars PBW 343 (carries *Yr9* and *Yr27*), PBW 621 (carries *Yr17*) and HD 2967 (gene not known) were highly popular among the farmers after their release in 2011. But presently all three cultivars are highly susceptible to stripe rust at seedling as well as at adult plant stages as their resistance has been broken down due to emergence of new pathotypes of the pathogen (110S119, 238S119). In previous study, the crosses of PBW 621 with PBW 343 and HD 2967 and evaluation of further generations (up to $F_4$) against pathotype 78S84 resulted in resistant segregants. In the present study, the $F_5$ and $F_6$ RIL populations have been evaluated against new pathotypes of *Pst*. The RILs categorized based on the disease severity on the P (Penultimate leaf) and F (flag) leaf into three categories i.e., high, moderate and low level of APR (adult plant resistance) having 1–200, 201–400 and >400 values of AUDPC, respectively, upon infection with stripe rust. The various APR components (latent period, lesion growth rate, spore production and uredial density) were studied on each category, i.e., resistant, moderately resistant and susceptible. The values of APR parameters decreased as the level of resistance increased. Based on molecular analysis, the lines (representing different categories of cross PBW 621 X PBW 343) containing the genes *Yr9* and *Yr17* due to their interactive effect provide resistance. Based on BSA using 35k SNPs and KASP markers association with phenotypic data of the RIL population (PBW 621 X HD 2967) showed the presence of two QTLs (*Q.Pst.pau-6B*, *Q.Pst.pau-5B*) responsible for the residual resistance and two SNPs AX-94891670 and AX-94454107 were found to be associated with the trait of interest on chromosome 6B and 5B respectively. The present study concludes that in the population of both the crosses (PBW 621 X PBW 343 and PBW 621 X HD 2967) major defeated gene contributed towards residual resistance by interacting with minor gene/QTLs.

**Funding:** The author(s) received no specific funding for this work.

**Competing interests:** The authors have declared that no competing interests exist.

## Introduction

Stripe or yellow rust of wheat caused by *Puccinia striiformis* f. sp. *tritici* has emerged as one of the most important disease throughout the world. It can cause huge losses in grain yields due to reduction in size and number of kernels [1]. Complete yield loss can occur, if infection occurs during very early in the cropping season [2]. Varied yield losses in the range of 10–70 per cent have been reported due to stripe rust in wheat growing areas of the world depending upon the earliness of initial infection, rate of disease development, duration of disease and susceptibility of cultivar [3]. Among all the available methods of stripe rust management, the cultivation of resistant cultivars is the most economical, effective and environment friendly. Till date, 83 permanently designated stripe rust resistance genes, 71 temporarily designated genes and 363 quantitative trait loci (QTLs) with different names have been reported in wheat [4–6]. Most of these genes are race/pathotype specific and confer all-stage resistance, which can be detected at the seedling stage, but a few are expressed only at later growth stages. Resistance based on a single major gene is often considered short lived due to genetic shifts in pathogen or the emergence of new virulence in the pathogen population in response to selection imposed by the host. Indeed, many stripe rust resistance genes have been short lived due to the emergence of new virulence in the pathogen or the increased prevalence of previously rare pathotypes [7]. The pathogen continues to evolve with respect to virulence (degree of pathogenicity of a particular pathotype/race of the pathogen) and aggressiveness (relative ability of a virulent isolate to cause disease on a susceptible host plant), often rendering resistant varieties susceptible. Consequently, emphasis has been placed on the identification and deployment of new sources of resistance and enhancing durability of resistance [3]. Collectively, several minor genes with additive effect can provide potentially durable resistance.

Wheat cultivar PBW 343, an Attila sib (ND/VG9144//KAL/BB/3/YACO/4/VEE#5), is a selection made at Punjab Agricultural University, Ludhiana, Punjab, India, from a set of lines called "Veery wheat derivatives" developed at CIMMYT, Mexico. After its release in the North Western Plain Zone (NWPZ) of India in 1995, PBW 343 emerged as a mega cultivar for more than a decade. By 2002–03, PBW 343 occupied more than 90 per cent of the wheat-growing area in Punjab and about 7 million hectares across the Indo-Gangetic Plains of India [8]. Apart from carrying multiple disease resistance, the 1B/1R wheat-rye translocation harboured by PBW 343 also possibly enhanced its adaptation zone and productivity [9]. By virtue of stripe rust-resistance gene *Yr27*, probably derived from Selkirk in its parentage, PBW 343 withstood the spread of *Yr9* virulence (46S119) (pathogenicity of virulent race/pathotype of the stripe rust pathogen resulting in breakdown of *Yr9* resistance), to which many other "Veery" derivatives succumbed [10,11]. The spread and increasing adaptation of *Yr27* virulence (development of virulence in the pathogen against the *Yr27* gene in wheat) to local conditions became evident in the form of stripe rust assuming damaging proportions in 2007 across an extensive wheat belt adjoining the Himalayan foothills [12]. The emergence of *Yr27* virulence (78S84) by the pathogen was followed by a step wise escalation in aggressiveness (in terms of adaptability) of the pathogen, as evidenced by the rapid decline in resistance in a set of post-*Yr27* gene wheat releases (e.g., PBW 550 (2005), DBW 17 (2007), PBW 621 (2011), and HD 2967 (2011)). In the backdrop of these evolutionary changes in the pathogen, newer wheat cultivars, viz., PBW 621 also known as 'KACHU' (KAUZ//ALTAR84/AOS/3/MILAN/KAUZ/4/ HUITES) and HD 2967 (ALD/COC//URES/3/HD2160M/HD2278), which were released in 2011 for timely sowing under irrigated conditions of NWPZ of India, showed progressive loss of resistance to stripe rust. The gene *Yr17* individually had lost resistance by 2012 and was present in PBW621 [8], however, PBW621 and HD2967 were elite and showed resistance at that time, so a population to map the gene(s)/QTLs was initiated. The two populations were developed, R x

S type (PBW621 x PBW343) and one R x R type (PBW621 x HD2967) to validate and pyramid the resistant components as well as testing the allelic nature of the target loci. These populations were screened against the pathotype 78S84 and were found conferring two gene(s) [13,14]. By the time population was advanced to $F_6$, there was a shift in the pathogen virulence (110S119), so the parents and derived populations became susceptible. However, in each population a proportion of lines were still resistant at adult plant stage. The populations were studied for the presence of known genes with the hypothesis that, the already defeated genes (*Yr9*, *Yr17*, *Yr27*, unknown gene/QTL from HD2967) when pyramided together reduce the extent of susceptibility or confer resistance. To elucidate these points the current study was undertaken. The first objective of the study was to find that major defeated genes (ghost resistance) contribute to residue resistance in the segregants of cross PBW 621 x PBW 343 and also to confirm the durability of surviving host residual resistance against prevalent stripe rust pathotypes. Since the source of resistance in HD 2967 was not known and R segregants were present in low number, we therefore planned to move forward for mapping of QTLs. Thus, the second objective of this study was to map the QTLs involved in residual stripe rust resistance from RIL population of cross PBW 621 x HD 2967 based on bulk segregant analysis using SNPs.

## Materials and methods

### Plant material

The plant material consisted of populations derived from crosses of cultivars PBW 621 (Pedigree: KAUZ//ALTAR84/AOS/3/MILAN/KAUZ/4/HUITES) with HD 2967 (Pedigree: ALD/COC//URES/3/HD2160M/HD2278) and PBW 343 (Pedigree: ND/VG9144//KAL/BB/3/YACO"S"/4/VEE#5). The derived populations were categorized previously by reaction against pathotype 78S84 during the crop seasons 2013–2016 by Singh et al., [13,14].

The RIL Population (690 $F_7$ RILs) and extremely resistant and susceptible phenotypic categories (84 $F_4$), of the cross PBW 621 × HD 2967 and extreme resistant and susceptible phenotypic categories (40, $F_4$) of the cross PBW 621 X PBW 343 were evaluated in the present study against the mixture of prevalent pathotypes of *P. striiformis tritici* during 2016–2018 (Fig 1). Each extreme phenotypic category consisted of 5 lines. During the selection, extreme phenotypic categories showing variable reaction for disease resistance in cropping season 2016–17 were rejected and the selected ones were further evaluated in the following season (2017–18). Moreover, resistant and susceptible extremes were selected based on the APR reaction on P leaf (penultimate leaf/one leaf below flag) and F (Flag) leaf (as all the planting material was susceptible at seedling and vegetative stage). Twenty phenotypic categories were selected from the cross PBW 621 × PBW 343 and sixty-one were selected from the cross PBW 621 × HD2967 for evaluation during season 2017–18. Along with these selected phenotypic categories, all 690 RILs of cross PBW 621 × HD2967 were evaluated against stripe rust during 2016–17 and only 200 randomly selected RILs were screened during 2017–18.

### Multiplication of inoculum

The inoculum of the prevalent pathotypes in Punjab (78S84, 46S119, 110S119 & 238S119) was procured from Regional Station, Indian Institute of wheat & Barley Research (IIWBR), Flowerdale, Shimla. The virulence/avirulence profile of the pathotypes is given in the S1 Table. The individual pathotype was first multiplied on susceptible cultivars namely Agra Local and PBW 343 under controlled conditions. The freshly produced uredospores of all the four pathotypes were mixed in equal proportion for further inoculations in field as well as under greenhouse conditions.

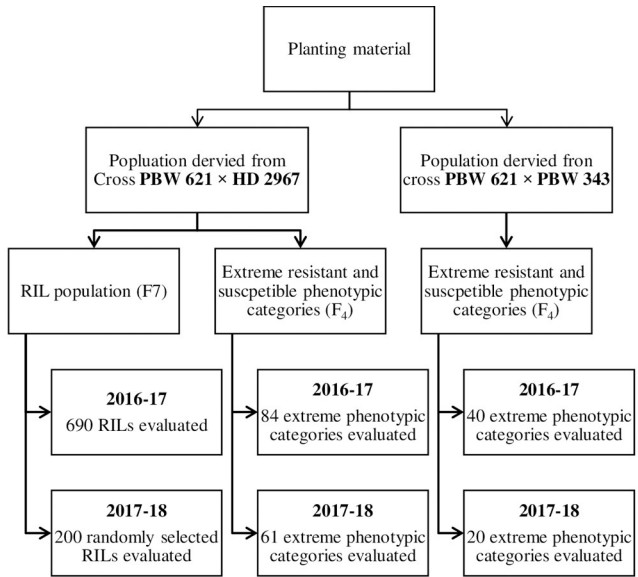

Note: During 2016-17 all the planting material evaluated at seedling (grrenhouse conditions) and adult plant stage (field conditions).
During 2017-18 evaluation was done under field conditions only.

**Fig 1. Details of planting material.**

## Evaluation against stripe rust

**At the seedling stage.** The extreme categories of both the crosses and 690 RILs of cross
PBW 621 × HD 2967 were also screened at seedling stage during cropping season 2016–17. All
the planting material was sown in 9-cm earthen pots containing potting mixture comprising
of coco-peat, vermiculite and soil in an equal ratio in duplicate. Parental cultivars and suscepti-
ble checks (Agra Local, PBW343 and A-9-30-1) were also sown. A single application of nitrog-
enous fertilizer urea was applied to seven-day-old seedlings. Seedlings were inoculated with
mixture of pathotypes i.e., 78S84, 46S119, 110S119 & 238S119 at 1–2 leaf stage by uredinio-
spores suspended in a light mineral oil using a propellant pressure. Inoculated seedlings were
incubated overnight at ambient temperature in a dark room in which mist was generated by a
humidifier. Following the dew treatment, infected seedlings were transferred to a greenhouse
growth room in which temperature was maintained within the range 10–15˚C. Disease assess-
ments were made 14 days after inoculation, and by following the 0–4 infection type (IT) scor-
ing system [15], in which IT '0' indicated no visible symptoms, IT ';' indicated hypersensitive
flecks, IT '1' indicated small uredinia with necrosis, IT '2' indicated small to medium-sized
uredinia with green islands and surrounded by necrosis or chlorosis, IT '3' indicated medium
to large sized uredinia with chlorosis, IT '4' indicated large uredinia without chlorosis, and IT
'X indicated heterogeneous ITs, similarly distributed over a given leaf. Plus and minus signs
were used to indicate variation in ITs, and the letters 'C' and 'N' were used to indicate more
than normal chlorosis or necrosis, respectively. Infection types of 3 or higher were regarded as
compatible (high infection type; high IT), whereas ITs less than 3 were regarded as incompati-
ble (low infection type; low IT). The presence or absence of genes in the test cultivars was pos-
tulated by correlating their responses to the array of pathotypes with those of control
differential genotypes. A high IT on the test cultivar indicated that it did not have any resis-
tance gene for which the test pathotype was avirulent.

**At the adult plant stage.** For evaluation at the adult-plant stage during both cropping sea-
sons 2016–17 and 2017–18, all the test material was grown in paired rows of one-meter row

length, with row to row spacing of 22.5 cm along with PBW 621, HD 2967 and PBW 343 as parental checks. A mixture of varieties consisting of PBW 343, Agra local, C-306, A-9-30-1 was used as an infector and spreader after every 20 rows and at the periphery of the plots for development of artificial epiphytotic conditions of stripe rust. For field inoculations, a mix of urediniospores, suspended in 10 litres of water with two drops of Tween 20, were sprayed during end December to early January using an ultra-low volume applicator on clear evenings with good expectation of dew to ensure good disease infection. Along with the sprays, pots containing infected plants were also kept in the experimental field to create homogeneous epiphytotic conditions.

Disease severity under field conditions was assessed at the adult plant stage according to the modified Cobb's scale. The recording process relies upon visual observations and the following intervals were used: Traces (TS), 5, 10, 20, 40, 60 and 100 per cent infection. The assessment of disease severity based on plant response was done according to modified Cobbs scale given by Peterson et al., [16]. Three successive disease severity observations on P and F leaves were recorded at 10-day intervals starting from the second week of February. From these observations various epidemiological parameters like AUDPC (area under the disease progress curve), rAUDPC (relative area under the disease progress curve), FRS (final rust severity) and CI (coefficient of infection) were estimated by taking mean values of disease reaction for P and F leaves.

Area under the disease progress curve (AUDPC): The AUDPC was calculated using the formula suggested by Wilcoxson et al., [17].

$$AUDPC = \sum_{i=1}^{k} 1/2(Si + Si - 1) \times d$$

(Where, Si = Disease severity at the end of time I, Si-1 = Disease severity at the end of time i-1, k = Number of evaluations of disease and, d = Interval between two evaluations.)

Relative area the under-disease progress curve (rAUDPC):

$$rAUDPC = \frac{AUDPC \; of \; line \; or \; variety}{AUDPC \; of \; infector} \times 100$$

Coefficient of infection (CI): CI is calculated by multiplying disease severity (DS) and constant values of infection type (IT). The constant values for infection types were used based on Immune = 0, R = 0.2, MR = 0.4, M = 0.6, MS = 0.8, S = 1 [18].

r Infection rate (r): The rate of rust infection was computed as described by Vanderplank [19]. The apparent rate of infection ('r') at different intervals was calculated by using the formula given by Vanderplank [19].

$$r = \frac{2.3}{t_2 - t_1} log_{10} \frac{x_2(1 - x_1)}{x_1(1 - x_2)}$$

(Where, r = apparent rate of growth of disease or rate of infection (units day$^{-1}$), $x_1$ = percent disease severity at $t_1$ date, $x_2$ = percent disease severity at date $t_2$)

## Genetic analysis of the disease data for the RIL population of cross PBW 621×HD 2967

Mendelian genetic analysis was carried out based on the distribution of phenotypic disease data into different categories for RIL population during the crop season 2016–17. The Chi-square test was used to determine the goodness of fit of the observed numbers of plants or

lines to the predicted segregation ratios of the RIL population to establish the number of stripe rust resistance genes and mode of inheritance.

## Estimation of APR components (Latent period, maximum lesion length, lesion growth rate, spore production and uredial density)

The extreme phenotypic categories derived from two crosses were divided into different classes susceptible (S), moderately resistant (MR) and resistant (R) based on AUDPC value (R: 1–200, MR: 201–400 and S: >400) in season 2016–17. During 2017–18, six categories were chosen from both the crosses (one each from class S, MR and R) to estimate the various APR components i.e., latent period, maximum lesion length, lesion growth rate, spore production and uredial density) in a green house. The test material (6 selected categories) was grown in pots and for each category three plants were maintained in each pot with 5 replications. The inoculations of P and F leaves were done after their emergence in February with a uredospore suspension ($2 \times 10^5$ spores/ml) in water and Tween-20 with camel hair brush onto the leaves.

Latent Period: Number of days from inoculation to first appearance of sporulating pustules was recorded.

Maximum lesion length and Lesion Growth rate: Lesion length was recorded after the appearance of pustules up to 8 days and lesion growth rate per day was worked out.

Uredial density: Photographs of pustules were taken after their appearance (before burst open of uredia) with a digital camera. Later the number of uredia was counted in marked areas. The area was measured with the help of online software SketchAndCalc™ (https://www.sketchandcalc.com/area-calculator).

Spore production: For counting the number of spores, measured bits of infected areas were dipped in water to suspend the spores. Number of spores was counted with the help of a haemocytometer.

## Molecular analysis

**Population PBW 621 x PBW 343.** For molecular analysis the DNA was isolated from the young leaves of six samples (two samples representing each category i.e. resistant, moderately resistant and susceptible of cross PBW 621 × PBW 343) and three parents (PBW 343, HD 2967 and PBW 621) using the standard CTAB (Cetyltrimethyl ammonium bromide) procedure given by Saghai-Maroof et al., [20]. DNA quantity and quality were assessed by using a NanoDrop™ 1000 spectrophotometer (ThermoScientific, Wilmington, USA). Quality and quantity of DNA was also checked by 0.8 percent agarose gel electrophoresis. Three markers for the genes *Yr9* (Xgwm582-1B; [21]), *Yr17* (Ventriup, LN-2 and *Yr17neg*; [22]) and *Yr27* (Xcdo405-2B/Xbcd152-2B; [10]) (S2 Table) were used to confirm the presence or absence of these genes. *In vitro* amplification using polymerase chain reaction (PCR) was performed in a 96-well PCR plate (Thermo Fischer Inc) in Eppendorf Master Cycler ProS. PCR amplification was carried out in a final reaction volume of 20μl, containing 2 mM MgCl₂, 1unit of Taq DNA polymerase, 1 mM dNTPs, 1x PCR buffer, 2μl of each marker and 30 ng of genomic DNA. The amplified products were electrophoresed in 2.5% agarose gels in 0.5 X TBE buffer, stained with good view nucleic acid stain (SBS genetech) at a concentration of 5.0 μl/100 ml of buffer. The gels were visualized under UV light and photographed using a Protein simple gel documentation system with the GeneSnap software programme.

**Single nucleotide polymorphism (SNP) and KASP genotyping of population PBW 621 x HD 2967.** *DNA extraction.* DNA was extracted from 10 resistant and 10 susceptible plants representing different extreme phenotypic categories of cross PBW 621 × HD 2967 and from

the randomly selected 200 RIL lines of the same cross by using the CTAB method. The quantity and quality of the DNA was checked by agarose gel electrophoresis and also by nano-quant. The DNA was stored at -20˚C until further use.

*BSA using Single nucleotide polymorphism (SNP) analysis.* The resistant bulk (RB) and susceptible bulk (SB) were prepared by mixing equal amounts of DNA from 10 resistant and 10 susceptible plants of extreme categories from cross PBW 621×HD 2967. The two bulks and parents were genotyped with the 35K SNP arrays. The sequences of all SNP markers were used to search the IWGSC RefSeq v 0.4 (http://www.wheatgenome.org/) to determine their physical positions using BLAST.

*Association of KASP markers with disease resistance.* Based on the genotypic data the eighteen SNPs located on the chromosomes 6A, 6B and 5B were selected. To identify the marker trait association between the markers found to be associated using BSA and the disease resistance in the population, the selected SNP markers were used to design Kompetitive Allele Specific PCR (KASP) markers (from the site http://polymarker.tgac.ac.uk/). The primers carrying standard FAM or HEX compatible tails (FAM tail: 5' GAAGGTGACCAAGTTCATGCT 3'; HEX tail: 5' GAAGGTCGGAG TCAACGGATT 3') with the target SNP at the 3' end were used. The reaction mixture for KASP was comprised of a final volume of 5 µl containing, 2.5 µl of 2× KASP V4.0 Mastermix, 0.056 µl of assay primer mix (12 mM of each allele-specific primer and 30 mM of the common primer) and 20–40 ng of genomic DNA. The reaction was performed in a Thermal Cycler with the cycling conditions: 94˚C for 15 min, 9 cycles of 94˚C for 20 sec, touchdown starting at 65˚C for 60 sec (decreasing 0.8 per cycle), 32 cycles of 94˚C for 20 sec, and 57˚C for 60 sec. Fluorescence was detected at ambient temperature. Data analysis was performed manually using Klustercaller software (version 2.22.0.5; LGC Hoddesdon, UK). The eighteen KASP markers were primarily applied on the parental genotypes to identify their usefulness based on amplification leading to clear clusters of the genotypes which were used for further analysis. The seven KASP markers (Table 1) were used on a random subset of the population for genotyping (200 RIL population). The genotypic data and phenotypic BLUEs were used to identify association of KASP markers to disease resistance (p-value <0.05) with significant difference testing using Kruskal-Wallis test. The effect size, based on H-statistic, was calculated by squared eta using the equation

$$eta^2 = (H - k + 1)/(n - k)$$

where H is the value obtained in the Kruskal-Wallis test, k is the number of alternate alleles, n is the total number of observations [23]. The effects are characterized as small effect (1% to < 6%), moderate effect (6% to < 14%), and large effect (> = 14%).

*Postulation of candidate genes.* In order to identify possible candidate genes of resistance for the identified QTLs, a region of 500kb on either side of the QTLs was scanned. All the high confidence genes as per the refseq V1.0 annotation were fetched, and their functions were

**Table 1. KASP markers applied on the RIL population of cross PBW 621 × HD 2967.**

| Primer name | FAM | HEX | Common |
|---|---|---|---|
| BA00330202 | GAAGGTCGGAGTCAACGGATTaccaccaacaaaagcttccA | GAAGGTGACCAAGTTCATGCTaccaccaacaaaagcttccG | cgcagataaagcagagagattG |
| BA00398753 | GAAGGTCGGAGTCAACGGATTccatccccttctcgctgttC | GAAGGTGACCAAGTTCATGCTccatccccttctcgctgttT | atctggtttgtgcggacc |
| BA00525667 | GAAGGTCGGAGTCAACGGATTtagtgtgtggtcatctggcA | GAAGGTGACCAAGTTCATGCTtagtgtgtggtcatctggcG | Gcacaactatgaagccaggt |
| BA00428050 | GAAGGTCGGAGTCAACGGATTagttctGaagctgaAgtgcA | GAAGGTGACCAAGTTCATGCTagttctGaagctgaAgtgcG | gcttgtgttcatgaacctgtaaatC |
| BA00855080 | GAAGGTCGGAGTCAACGGATTgTtGAaCgtgctCacctgC | GAAGGTGACCAAGTTCATGCTgTtGAaCgtgctCacctgT | cgaagtaggtTtcgccgtaG |
| BA00402838 | GAAGGTCGGAGTCAACGGATTcctttgcagcgaggcgaT | GAAGGTGACCAAGTTCATGCTcctttgcagcgaggcgaC | aAagagctcacccatgcC |
| BA00503975 | GAAGGTCGGAGTCAACGGATTcggggtacagcaaggtCC | GAAGGTGACCAAGTTCATGCTcggggtacagcaaggtCT | tgccCatggaagctggatC |

identified. These functions were then literature searched to target/shortlist the genes with possible role into disease resistance.

## Results

### Evaluation of parents and population against stripe rust

All three parents and the population when inoculated at the seedling stage under controlled conditions with a mixture of prevalent pathotypes (78S84, 110S119, 46S119 and 238S119) showed IT of 4 i.e., they were completely susceptible at seedling stage. The three parents PBW 621, PBW 343 and HD 2967 were also susceptible against each pathotype individually. At the vegetative stage (under field conditions), all three parents along with the RIL population were susceptible (60-80S). To determine the presence of APR, the disease scoring was done on P (Penultimate leaf/one leaf below flag leaf) and F (Flag) leaf. The disease score of three parents PBW 343, HD 2967 and PBW 621 at adult plant stage was 80-100S, 60-80S and 40-60S, respectively. During 2016–17, all the extreme categories were evaluated, but in 2017–18 only selected categories were evaluated. The phenotypic categories showing variable reaction for disease resistance during cropping season 2016–17 were rejected. For cross PBW 621 × PBW 343, out of 40 categories evaluated in 2016–17, 20 showed variation for disease resistance and were rejected. Similarly, for cross PBW 621 × HD 2967, out of 84 sets, 23 were rejected. Data regarding the disease severity and mean values of epidemiological parameters (AUDPC, rAUDPC, FRS and CI) of different selected extreme phenotypic categories (not showing variable reaction) for both the seasons is presented in the S3 and S4 Tables. All the epidemiological parameters were calculated by taking mean values of disease severity on P and F leaves. Based on the AUDPC, the extreme phenotypic categories were grouped into three classes, i.e., high, moderate and low levels of APR having 1–200, 201–400 and >400 values of AUDPC, respectively. In extreme phenotypic categories of cross PBW 621 × PBW 343, out of 20 extreme categories, 9 had a high level of APR, 10 had a low level of APR and one set (9) showed a different reaction in both the years (Table 2). The range of other epidemiological parameters (rAUDPC, FRS and CI) is given for each category of APR. Extreme phenotypic category 9 showed a moderate level of APR during 2016–17 and high in 2017–18. Similarly, the extreme categories of cross PBW 621 × HD 2967 were also categorized based on the

**Table 2. Categorization of extreme phenotypic categories of 2 crosses (PBW 621× PBW 343 and PBW 621 × HD 2967) based on AUDPC* (2016–17 and 2017–18).**

| Cross | AUDPC* | Level of APR* | Number of extreme categories | rAUDPC* | FRS* | CI* |
|---|---|---|---|---|---|---|
| **PBW 621× PBW 343** | 1–200 | High | 10 | 3.8–14.2 | 5–15 | 4–15 |
| | 201–400 | Moderate | 1 | 18.7 | 22.5 | 22.5 |
| | >400 | Low | 10 | 36.3–75.2 | 35–60 | 35–60 |
| | **Note: Lines of one category showed variable disease reaction in both years** | | | | | |
| **PBW 621× HD 2967** | 1–200 | High | 40 | 1–10 | 2.5–12.5 | 2–12.5 |
| | 201–400 | Moderate | 5 | 17.3–29.8 | 20–30 | 20–30 |
| | >400 | Low | 13 | 32.2–75.2 | 32.5–60 | 32.5–60 |
| | **Note: lines of 3 categories showed variable disease reaction in both the years** | | | | | |

*AUDPC–Area under the disease progress curve, *APR–Adult plant resistance, *rAUDPC–Relative area under the disease progress curve, *FRS–Final rust severity, *CI–Coefficient of infection.

AUDPC (Table 2). Out of 61 extreme phenotypic categories of PBW 621 × HD 2967, 40 were having high, 5 moderate, 13 low level of APR and the remaining 3 categories showed variable responses in both the years.

## APR components study

The extreme phenotypic categories derived from two crosses were divided into different classes (Susceptible (S), moderately resistant (MR) and resistant (R)) based on AUDPC values in 2016–17. Six categories were selected, and the various APR components were estimated on these selected categories as well as on the parents (PBW 621, PBW 343 and HD 2967) during 2017–18. The various parameters of APR, i.e., latent period, maximum lesion length, lesion growth rate, spore production and uredial density are given in Table 3.

The latent period on different categories of two crosses and on the parents (PBW 621, PBW 343 and HD 2967) varied from 14–16 days. The other four parameters maximum lesion length, lesion growth rate, spore production and uredial density, showed significant difference within each category.

The maximum lesion length (3.20), lesion growth rate (0.37), spore production ($1.72 \times 10^5$) and uredial density (27.12) were highest on susceptible category as compared to the other two

**Table 3. Adult plant resistance components on selected extreme phenotypic categories from 2 crosses (PBW 621 X PBW 343 and PBW 621 X HD 2967).**

| Extreme phenotypic category | APR category* (Rust severity) | Category No. | Latent Period | Maximum lesion length (cm) | Lesion growth rate/day | Spore production/ cm² | Uredial density/ cm² |
|---|---|---|---|---|---|---|---|
| **PBW621×PBW343** | Susceptible (60S) | 1 | 14 | 3.20 | 0.37±0.01 | $1.72 \times 10^5 \pm$ 3219.06 | 27.12 ±0.82 |
| | Moderately resistant (5-10MS) | 11 | 15 | 2.62 | 0.30±0.01 | $5.80 \times 10^4 \pm$ 1565.99 | 21.25 ±0.50 |
| | Resistant (5S) | 7 | 16 | 1.48 | 0.17±0.02 | $2.09 \times 10^4 \pm$ 1690.20 | 17.05 ±0.40 |
| Parent | PBW 621 (40S) | | 14 | 4.34 | 0.21±0.01 | $1.67 \times 10^5 \pm$ 3440.28 | 31.82 ±0.81 |
| | PBW 343(80S) | | 14 | 4.60 | 0.32±0.01 | $3.22 \times 10^5 \pm$ 2484.17 | 41.64 ±1.38 |
| LSD** | | | - | 0.85 | 0.04 | 7656.29 | 2.52 |
| **PBW621×HD2967** | Susceptible (60S) | 3 | 14 | 4.04 | 0.35±0.02 | $1.40 \times 10^5 \pm$ 6292.48 | 32.90 ±0.96 |
| | Moderately Resistant (5-10MS) | 58 | 15 | 1.96 | 0.15±0.01 | $5.64 \times 10^4 \pm$ 2921.01 | 20.38 ±0.83 |
| | Resistant (5S) | 2 | 15 | 1.86 | 0.14±0.01 | $3.91 \times 10^4 \pm$ 1872.65 | 17.92 ±0.54 |
| Parent | PBW 621 (40S) | | 14 | 4.34 | 0.21±0.01 | $1.67 \times 10^5 \pm$ 3440.28 | 31.82 ±0.81 |
| | HD 2967 (60S) | | 14 | 4.50 | 0.29±0.02 | $1.64 \times 10^5 \pm$ 2291.70 | 36.30 ±0.51 |
| LSD** | | | - | 0.96 | 0.05 | 10936.83 | 2.21 |

*Disease reaction in the field,

**LSD- Least Significant difference.

categories (MR and R) in case of cross PBW 621 × PBW 343. The values for these four parameters decrease as the level of resistance increases. The maximum lesion length, uredial density, and spore production were higher on the parents in comparison to the susceptible category while lesion growth rate was low.

In case of cross PBW 621× HD 2967 lesion growth rate, spore production and uredial density were maximum on susceptible category having values of 4.04, 0.35, $1.40×10^5$ and 32.90, respectively. These three parameters decrease as the level of resistance increases from moderately resistant to resistant and are minimum in the resistant category. The maximum lesion length, lesion growth rate, spore production and uredial density were more on the parents (PBW 621 and HD 2967) than the resistant and moderately resistant categories. Comparison of the susceptible category with both parents revealed that spore production was less, and lesion growth rate was more in the susceptible category. The maximum lesion length was at par with both the parents. The mean uredial density in the susceptible category was less than that for HD 2967 but at par with PBW 621.

Overall, these results revealed that three parameters, i.e., lesion growth rate, spore production and uredial density were less in the resistant category as compared to the parents (PBW 621, PBW 343 and HD 2967). Although the parents were susceptible, recombination of these susceptible lines gave rise to new assemblage of host resistance components. The additive effect of minor genes/QTLs showed the resistance in segregating generations of both crosses. Singh et al., [14] hypothesised that in the cross PBW 621×PBW 343 the resistant segregants possessed two genes, one contributed by PBW 621 and the other, unexpectedly but obviously, came from the most susceptible cultivar, PBW 343. This hypothesis is confirmed by the APR components study and for further validation of this hypothesis for cross PBW 621 × PBW 343 molecular analysis was done for three genes *Yr9*, *Yr17* and *Yr27*. For other cross i.e., PBW 621 × HD 2967, since the source of resistance was not known in HD 2967, so we proceed for QTL mapping.

## Molecular confirmation of residual resistance in the population of cross PBW 621 × PBW 343

In the cross PBW 621 × PBW 343, the presence of residual resistance was determined by applying the markers for gene *Yr9*, *Yr17* and *Yr27* (S2 Table) on DNA of single plants selected from different phenotypic categories of cross PBW 621 × PBW 343. The markers for these three major genes were selected to confirm the role of defeated genes in imparting additive residual resistance. The amplification pattern of these genes is presented in Table 4 and Fig 2. In the susceptible category *Yr9* and *Yr27* were present and *Yr17* was negative, while in the moderately resistant and resistant categories *Yr9* and *Yr17* were present and *Yr27* was absent. The two

**Table 4. Amplification pattern of markers of genes *Yr9*, *Yr17* negative and *Yr27* applied on different categories of resistant selected from population of cross PBW 621 × PBW 343.**

| Category of resistance | Yr9 | Yr17 negative | Yr27 |
|---|---|---|---|
| Susceptible (S)[*] | + | - | + |
| Moderately resistant (MR) | + | + | - |
| Resistant (R) | + | + | - |
| PBW 343 | + | - | + |
| PBW 621 | - | + | - |
| HD 2967 | - | - | - |

[*]DNA was taken from single plant representing each category i.e., Susceptible (S), Moderately Resistant (MR) and Resistant (R) of cross PBW 621 × PBW 343.

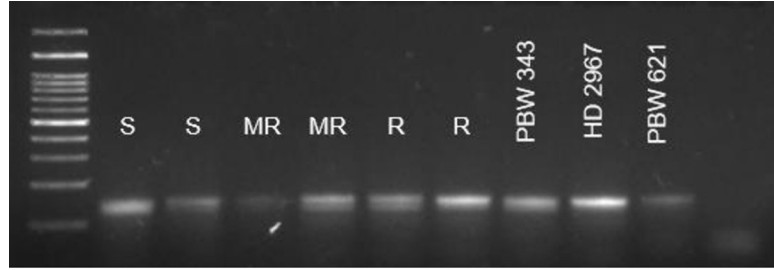

*Yr9* (**Xgwm582-1B**)

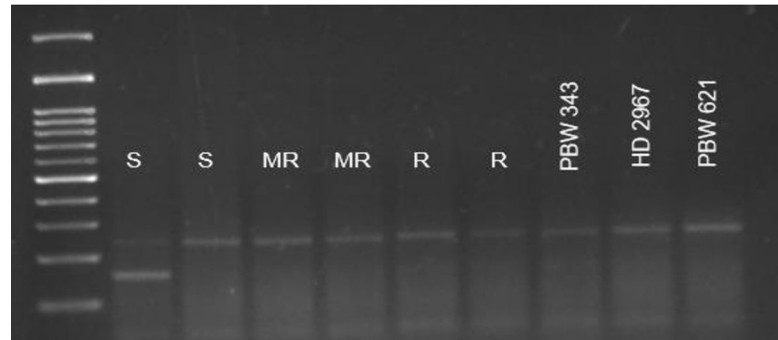

*Yr17* (**VENTRIUP/LN2/***Yr17neg***)

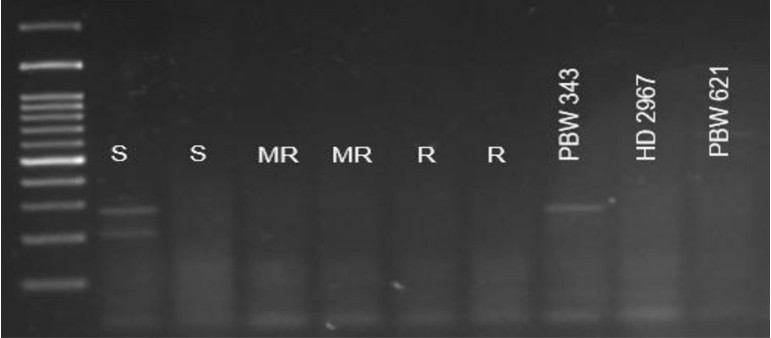

*Yr27* (**Xcdo405-2B/Xbcd152-2B**)

**Fig 2. PCR amplification products for the samples of different categories of resistance along with the parents obtained by applying gene specific primers (Lane I contains a 100bp ladder).**

genes (*Yr9* and *Yr27*) were present in parent PBW 343, while parent PBW 621 carries only one gene (*Yr17*). Thus, assemblage of two genes *Yr9* and *Yr17* provides MR-R type reaction at the adult plant stage in moderately resistant and resistant categories of cross PBW 621 × PBW 343. The gene *Yr9* was contributed by parent PBW 343 and *Yr17* comes from PBW 621. The major defeated gene *Yr9* from susceptible parent PBW 343 contributed towards the resistance by interacting with gene *Yr17*.

## Genetic analysis of the RIL population (F$_8$ generation) of cross PBW 621×HD 2967 against stripe rust

The 690 lines of the RIL population from cross PBW 621 X HD 2967 were evaluated during the season 2016–17 under field conditions against stripe rust (mixture of pathotypes) at adult

**Table 5. Segregation of $F_8$ RIL population of cross PBW 621 X HD 2967 against stripe rust mixture pathotype.**

| Phenotypic Classes | Genotype | Expected ratio of plants in classes | RIL population of cross PBW621 x HD 2967 | | |
|---|---|---|---|---|---|
| | | | Observed | Expected | Chi-square value |
| *Highly Resistant | AABBCC | 1 | 76 | 86 | 1.16 |
| Moderately Resistant and Moderately Susceptible | AABBcc, AAbbCC, aaBBCC, AAbbcc, aaBBcc, aabbCC | 6 | 510 | 518 | 0.12 |
| *Highly Susceptible | Aabbcc | 1 | 104 | 86 | 3.77 |
| | Total | | 690 | | 5.05 |

*Disease score of parents: PBW 621–40-60S, HD 2967–60-80S.

*Highly Resistant- disease score ≤5S.

*Highly Susceptible- disease score ≥60S.

Gene and ratio postulation: 3, 1:3:3:1, where intermediates are pooled and ratio apparent as 1:6:1.

plant stage. The segregation pattern for stripe rust resistance in $F_8$ generation of the RIL population is presented in Table 5. The RIL population was divided into 3 classes; class-I consisting of highly resistant lines (≤5S), Class-II having moderately resistant and moderately susceptible lines (10S, 20S, 40S) and class-III contained highly susceptible lines (≥60S). Class-I had 76 lines, class-II contained 510 lines and class-III had 104 lines. The chi-square analysis was done, and an acceptable value of chi square ($\chi^2_{1:6:1}$ = 5.05, p-value = 0.08, df = 2) was obtained, indicating three genes involved in additive manner contributed towards resistance. The genetic analysis of earlier generations ($F_2$ and $F_3$) showed a simple situation of inheritance of resistance in which each parent was contributing one resistance gene which were further seen to combine in an additive manner to give a transgressive resistant category [24].

## Genotyping with SNP and KASP assay for QTL mapping on population of cross PBW 621 × HD 2967

In the cross PBW 621 × HD 2967 the residual resistance may also be due to the additive effect of minor genes/QTLs, since the source of resistance in parent HD 2967 was not known. Therefore, for this cross we planned to move forward for mapping of QTLs involved in residual stripe rust resistance from population of cross PBW 621 x HD 2967 based on bulk segregant analysis using SNPs. The resistant and susceptible bulks (made by comprising equal amount of DNA from 10 resistant and 10 susceptible plants) were genotyped with 35K SNPs. A total of 87 SNPs showed polymorphism between the parents and two bulks. Out of these 87 SNPs, major clusters were formed on chromosome 6A and 6B. The KASP markers were designed from SNPs of these two chromosomes (6A and 6B) with different consensus positions. The chromosome 5B SNPs were also selected for KASP marker synthesis, based on the previous molecular tagging of stripe rust resistance on this chromosome by Singh [24]. A total of eighteen KASP markers were designed from the SNPs located on chromosomes 5B, 6A and 6B.

These eighteen KASP markers were first applied to the two parental genotypes (PBW 621 and HD 2967) to identify and validate the PCR amplification. Out of 18 markers identified using BSA study, only seven markers were found to be showing clear clusters in the KASP assay. The phenotypic BLUEs and genotypic data from these markers was used to identify association of the markers with the disease resistance by significant difference testing using

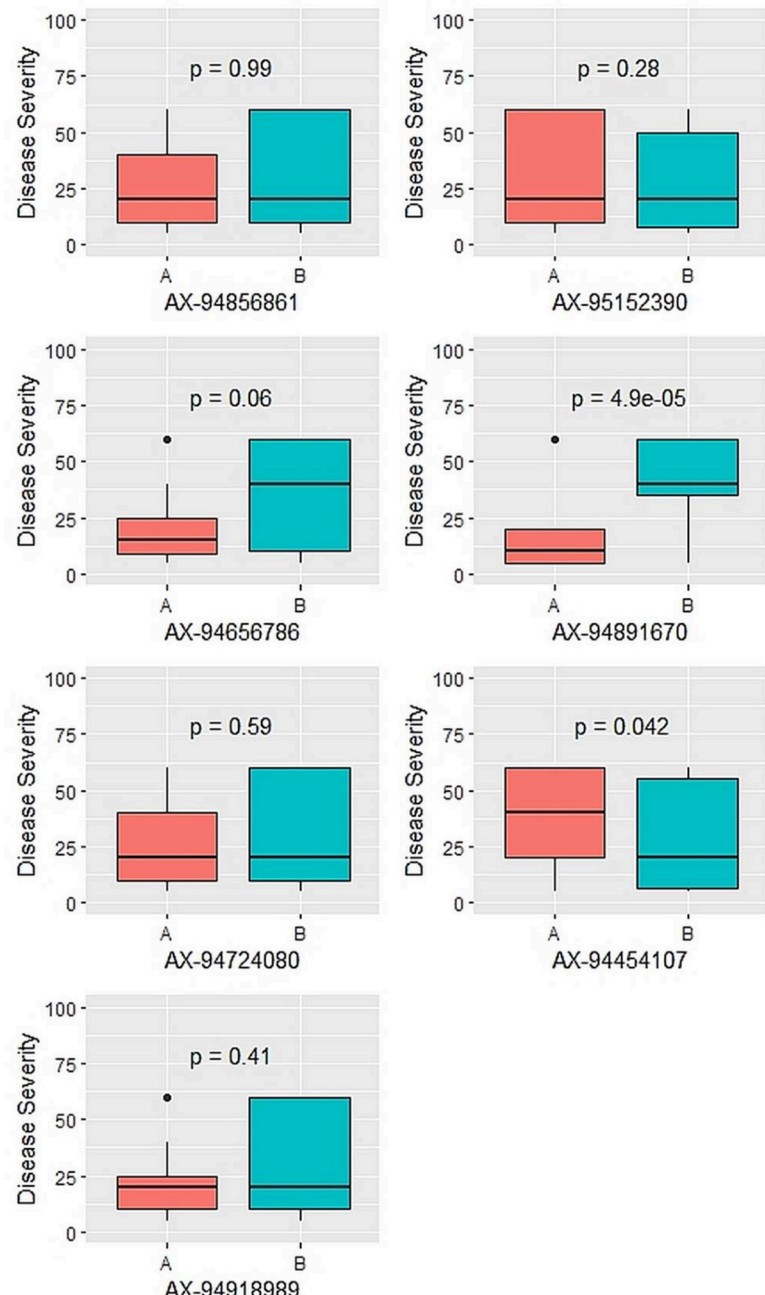

**Fig 3. Boxplots showing the effect of phenotypic variation between the two alleles of the SNP markers linked to QTLs for disease score of stripe rust.** Kruskal-Wallis test was used to determine the significant differences between the mean values of two alleles.

Kruskal-Wallis test. The significant difference testing identified two markers to be associated with disease resistance at p-value < 0.05. The marker AX-94891670 on chr6B at 39.766 Mb was found to be associated with disease resistance (p-value 4.90E-05) with resistance allele being donated by PBW621 (Fig 3, Table 6). The marker AX-94454107 on chr5B at 381.040 Mb was found to be associated with disease resistance (p-value 4.15E-02) with resistance allele being donated by HD2967. The effect size study of the two QTLs (*Q.Pst.pau-6B*, *Q.Pst.pau-5B*)

**Table 6. QTLs identified and their association with the KASP Markers based on significant difference by Kruskal-Wallis test.**

| QTL | Marker | Chr | Pos (Mb) | $H^2$ | p-value | effect(%) | mag. |
|---|---|---|---|---|---|---|---|
| *Q.Pst.pau-6B* | AX-94891670 | chr6B | 39.766 | 16.485 | 4.90E-05 | 21.81 | large |
| *Q.Pst.pau-5B* | AX-94454107 | chr5B | 381.040 | 04.155 | 4.15E-02 | 03.71 | small |

# chromosome (chr), H-statistic based on Kruskal-Wallis test (H2), magnitude (mag).

identified that marker AX-94891670 has large effect (21.81%) whereas marker AX-94454107 showed small effect (3.71%).

## Postulation of candidate genes

The physical locations of the SNPs linked to the QTLs detected in the present study were extracted from CerealsDB database [25]. The physical positions were used to identify the genes present adjacent to them in a region of 500 kb on either side of the SNP. For each target locus, the regions were inspected to identify candidate genes for the QTL and the genes known to be involved in different pathways of pathogen–host interactions and pathogenesis were considered to understand their role in imparting resistance to stripe rust (Table 7). The genomic region of the SNP AX-94454107 linked to QTL *Q.Pst.pau-5B* was found to be bearing five candidate genes *TraesCS5B01G210300*, *TraesCS5B01G210400*, *TraesCS5B01G210500*, *TraesCS5B01G210600*, and *TraesCS5B01G210700*. Similarly, the genomic region of the SNP AX-94891670 linked to QTL *Q.Pst.pau-6B* encompasses 11 candidate genes

**Table 7. Postulation of genes present in the survey sequence of wheat genome refseqV1.0.**

| QTL | SNP | Chr | Pos (Mb) | GeneID | Dist. SNP (Kb) | Function |
|---|---|---|---|---|---|---|
| *Q.Pst. pau-5B* | AX-94454107 | chr5B | 381.040 | *TraesCS5B01G210300* | 226.561 | Receptor-like protein kinase |
| | | | | *TraesCS5B01G210400* | 223.328 | DNA ligase |
| | | | | *TraesCS5B01G210500* | -1.898 | Early auxin response protein |
| | | | | *TraesCS5B01G210600* | -307.569 | Transmembrane protein 87A |
| | | | | *TraesCS5B01G210700* | -315.846 | Exocyst complex component, putative |
| *Q.Pst. pau-6B* | AX-94891670 | chr6B | 39.766 | *TraesCS6B01G059500* | 454.433 | Mitochondrial transcription termination factor-like |
| | | | | *TraesCS6B01G059600* | 439.367 | Mitochondrial transcription termination factor-like |
| | | | | *TraesCS6B01G059700* | 426.609 | Dirigent protein 17 |
| | | | | *TraesCS6B01G059800* | 398.479 | Receptor-like protein kinase |
| | | | | *TraesCS6B01G059900* | 309.683 | Leucine-rich repeat receptor-like protein kinase family protein |
| | | | | *TraesCS6B01G060000* | 89.796 | F-box family protein |
| | | | | *TraesCS6B01G060100* | 10.783 | Leucine-rich repeat receptor-like protein kinase family protein |
| | | | | *TraesCS6B01G060200* | 6.043 | NAD(P)-binding Rossmann-fold superfamily protein |
| | | | | *TraesCS6B01G060300* | -37.971 | Terpene synthase |
| | | | | *TraesCS6B01G060400* | -379.145 | F-box family protein |
| | | | | *TraesCS6B01G060500* | -486.012 | F-box family protein |

(*TraesCS6B01G059500*, *TraesCS6B01G059600*, *TraesCS6B01G059700*, *TraesCS6B01G059800*, *TraesCS6B01G059900*, *TraesCS6B01G060000*, *TraesCS6B01G060100*, *TraesCS6B01G060200*, *TraesCS6B01G060300*, *TraesCS6B01G060400*, *TraesCS6B01G060500*). The various categories of these genes on the mapped QTLs are receptor-like protein kinases (RLKs), DNA ligase, auxin response factors (ARFs), transmembrane protein, exocyst complex component, mitochondrial transcription termination factor-like (mTERF), Dirigent protein 17 (DIR), Leucine-rich repeat receptor-like protein kinase family protein, F-box family protein, NAD(P)-binding Rossmann-fold superfamily protein and terpene synthase (Table 7). All these genes are known for their action in imparting resistance through pathogen recognition, as well as being involved in various biotic and abiotic stresses or plant-pathogenesis pathways.

## Discussion

Stripe rust or yellow rust is the major obstacle in the cultivation of wheat. The use of resistant cultivars is the most effective and economical method for management of this disease. The resistance present in the cultivars can be major gene (seedling resistance) or minor gene (adult plant resistance). The resistance based on single gene is short lived and can be easily broken down. Consequently, there is always need of the new resistance sources with higher durability. The durable resistance can be achieved with the additive effect of minor genes/QTLs. The cumulative effect of minor gene(s)/QTL(s) and major defeated gene can also provide durable resistance which can be exploited in plant breeding for stripe rust resistance.

The results of the present study indicated that segregants from the crosses of PBW 621 with PBW 343 and HD 2967 possessed a significant level of resistance at adult plant stage. Both HD 2967 and PBW 621 were resistant initially, then showed a progressive loss of resistance over time due to change in pathogen population. Singh et al., [13] and Singh et al., [14] reported that the resistant segregants from each cross (PBW 621/PBW343 and PBW621/HD 2967) were hypothesized to carry two genes of resistance against stripe rust disease of wheat, one each from the resistant and susceptible parent. In the present study, this hypothesis was confirmed for the cross PBW 621 × PBW 343 by APR component study and with gene specific markers. The various APR components (lesion growth rate, spore production and uredial density) were also less in the moderately resistant and resistant categories due to the interactive effect of two genes (*Yr9* and *Yr27*). The major defeated gene *Yr9* from susceptible parent PBW 343 contributed towards the resistance by interacting with *Yr17*. Brodny et al., [26] also demonstrated that the lines with two or three defeated *Sr* genes show reduced pustule size and sporulation than lines with single genes and stated that each of the three resistance genes has a residual expression when confronted by matching virulence genes. The adult plant resistance lines in the moderately resistant and resistant category with residual resistance can be further used in breeding programs to enhance the durability of resistance against different prevalent rust pathotypes.

The surviving host residual resistance remained effective from 2012 to 2016 [13,14] and during present study (in cropping season 2016–17 and 2017–18), although some extreme phenotypic categories for two crosses showed variation for disease resistance in cropping season 2016–17 and were rejected. Thus, the surviving host residual resistance was more durable (effective for 6 years) than the one that was overcome by the pathogen in the previous years.

In cross PBW 621 × HD 2967, two QTL (*Q.Pst.pau-6B*, *Q.Pst.pau-5B*) were mapped which were found to contribute for residual resistance. Further study of the annotated reference of wheat genome refseqV1.0 showed the genes present in the genomic regions of two mapped QTLs. Various categories of these genes are receptor-like protein kinases (RLKs), DNA ligase, auxin response factors (ARFs), transmembrane protein, exocyst complex component,

mitochondrial transcription termination factor-like (mTERF), Dirigent protein 17 (DIR), Leucine-rich repeat receptor-like protein kinase family protein, F-box family protein, NAD(P)-binding Rossmann-fold superfamily protein and terpene synthase. All these are known for their action in pathogen recognition and being involved in various biotic and abiotic stresses or plant-pathogenesis pathways. The Receptor kinases are known to be modulating plant defence responses. Receptor-like kinases (RLKs) and receptor-like proteins (RLPs) act as pattern recognition receptors (PRRs) [27] and thus lead to first defence response. Multi-protein immune complexes of PRRs and other RLKs are formed at the surface of interaction. In wheat, *TaRLK-R1,2,3* [28] and LRK10 [29] have been involved in plant immunity where TaRLK-R1 has also been cloned [30].

The auxin response factors (ARFs) in various studies has been explained as a mediator of auxin to biotic and abiotic stresses [31–33]. Bouzroud et al., [33] reported that ARFs have a vital role in alteration (activation or repression) of the rate of transcription of auxin responsive genes. Both biotic and abiotic stress-responsive genes are enriched in cis-elements of 5′-regulatory units in ARFs. They showed that under stress conditions, ARFs are actively regulated at the post-transcriptional level.

The transmembrane protein regulates the fungal development and pathogenicity via the MAPK module [34]. The plant exocyst complex acts as an important component in the regulation of polarized cell growth; a role that might also strongly influence the capability of plant cells to react to pathogen attack. Exocyst subunits; Exo70B2 and Exo70H1 are involved in the response to pathogens, with Exo70B2 having a more important role in cell wall apposition formation related to plant defence [35].

The leucine-Rich Repeats Receptor-Like Kinases represent a large and complex gene family in plants, mainly involved in development and stress responses. The mitochondrial transcription termination factor-like (mTERFs) are best known to act against abiotic stresses and since only eight plant mTERFs are known to be characterized, very little is known about their action against biotic stresses [36–39].

The F-box proteins are involved in plant growth and development. For example, F-box protein-FOA1 involved in abscisic acid (ABA) signaling to affect the seed germination [40]. The Avr9/Cf-9 rapidly elicited (ACRE189)/Avr9/Cf-9–induced F-BOX1 (ACIF1) proteins can regulate cell death and defense when the pathogen is recognized in the Tobacco and Tomato plant [41]. The dirigent proteins (DIR) proteins also play essential roles in guiding the correct formation of lignin. Lignin is an important component of the cell wall and is essential in resistance to adverse external environments and provides a physical barrier for healthy plant growth [42,43]. The synthesis of terpenes by terpene synthase is one of the responses to attack in numerous plant-pathogen binomials, where terpenes act as specialized or generalized pathogen inhibitors [44]. The role of proteins NAD(P)-binding Rossmann-fold superfamily and DNA ligases in plant pathogenesis is either yet not reported or not well documented.

The present study showed that in both the crosses (PBW 621 × PBW 343 and PBW 621 × HD 2967) recombination of minor components of resistance in susceptible lines gave rise to new assemblage of host resistance components imparting residual resistance which seems durable. It's a general perception on the resistance in wheat varieties that the quantitative disease resistance (QDR) is more durable than the resistance provided by the major genes [45]. Major resistance genes usually provide high levels of resistance and are easy to mobilize into the desirable agronomic backgrounds but the major problem with these is the lack of durability, especially in case of a highly evolving pathogen, as in the case of *Pst* pathotypes. The ephemeral nature of this type of resistance and its subsequent breakdown has been well demonstrated during the appearance of pathotype 78S84 of stripe rust in northern India to which

the variety PBW343 succumbed completely followed by the plethora of susceptibility tales in all the cultivars of those times.

It has been observed that after the break down of major resistance genes QDR plays a significant role. QDR may be the result of weaker forms of major defeated genes especially when few are put together, hence appropriately often termed as ghost resistance. This has also been seen in the present study. In the resistant segregant of cross PBW 621 × PBW 343, major defeated genes *Yr9* and *Yr17* contributed towards resistance i.e., presence of ghost resistance was there. The simultaneous evolution of plant–host pathogen relationships may be governed by the stabilizing selection hypothesis as stated by Vanderplank in 1963 [46]. Later Leonard and Czochor [47] observed that the new aggressive and virulent pathotypes of pathogen usually suffer a loss in general fitness while acquiring new virulence. As a result, the more virulent pathogen will have lower fitness. Though it is difficult to obtain evidence for this, the concept and resistance from defeated residual genes or ghost genes supports this hypothesis. A residual effect on disease resistance in our study is highly correlated with pathogen fitness. Actually, the breakdown of resistance genes is associated with a penalty to fitness of stripe rust pathogen. Riley [48] gave the theory of ghost resistance stating that R genes for vertical resistance in the host plant are defeated by matching virulence gene in the pathogen, a ghost of resistance survives. This was further confirmed by experiments of Samborski and Dyck [49] on a backcross line developed from Neepwa wheat with the defeated leaf rust resistance genes *Lr11* and *Lr30*. Residual genes add to the complexity of the inheritance of stem rust resistance [26]. They studied the effect of genes *Sr6*, *Sr8* and *Sr9* against a race virulent on all these genes. They have observed that each gene has an effect on the pustule size and sporulation. Thus, the two-gene combinations are more effective. Pederson and Leath [50] also suggested the pyramiding of defeated genes to maintain their effects. Our observation that many RILs which are highly resistant to the prevalent virulent strain of *Pst* can be recovered from the cross of two susceptible parents. This leads to a major implication in breeding for disease resistance as a high level of durable resistance may be achieved by the cumulative effect of multiple minor QTLs combined with residual effects of defeated major genes. In practice, a breeding program for stripe rust resistance in wheat may not necessarily involve highly resistant parents as long as the resistance genes/QTLs in so-called susceptible cultivars are complementary to each other. A phenomenon well demonstrated in CIMMYT wheat breeding programme where APRs are pyramided with minor genes. The precise genomic advances for disease resistance versus susceptibility will produce a deeper understanding of plant defence mechanisms and ghost resistance to be utilized for potential benefits in wheat breeding programs.

## Supporting information

**S1 Table. Avirulence/virulence formula of predominant Indian wheat rust pathotypes used in the present study.**
(DOCX)

**S2 Table. Primer name, sequence of three stripe rust resistance genes (*Yr9, Yr17* and *Yr27*).**
(DOC)

**S3 Table. Disease reaction in field nursery and epidemiological parameters on extreme phenotypic categories of cross PBW 621× PBW 343.**
(DOC)

**S4 Table. Disease reaction in field nursery and epidemiological parameters on extreme phenotypic category of cross PBW 621× HD 2967.**
(DOC)

**S1 Raw images.**
(PDF)

# Acknowledgments

The first author (Harpreet Singh) acknowledges the fellowship provided during Ph.D. program by Department of Science & Technology, Govt. of India under 'INSPIRE fellowship scheme (IF 160704). The authors are thankful to Dr S C Bhardwaj, Director, Regional Station, IIWBR. Flowerdale, Shimla for providing the inoculum of *Puccinia striiformis* f. sp. *tritici.*

# Author Contributions

**Conceptualization:** Jaspal Kaur, Puja Srivastava, Navtej Singh Bains.

**Data curation:** Harpreet Singh, Jaspal Kaur, Puja Srivastava, Gomti Grover.

**Formal analysis:** Harpreet Singh, Jaspal Kaur, Guriqbal Singh Dhillon.

**Investigation:** Harpreet Singh, Jaspal Kaur, Ritu Bala, Puja Srivastava, Achla Sharma, Gomti Grover, Rupinder Pal Singh.

**Supervision:** Jaspal Kaur, Parveen Chhuneja, Navtej Singh Bains.

**Writing – original draft:** Harpreet Singh, Jaspal Kaur, Puja Srivastava.

**Writing – review & editing:** Harpreet Singh, Jaspal Kaur, Puja Srivastava, Guriqbal Singh Dhillon.

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
