## [Decision Letter · Decision Letter 0]

26 Nov 2021

PONE-D-21-31839Residual effect of defeated stripe rust resistance genes/QTLs in bread wheat against prevalent pathotypes of Puccinia striiformis f. sp. triticiPLOS ONE

Dear Dr. HARPREET SINGH,

Thank you for submitting your manuscript to PLOS ONE. After careful consideration, we feel that it has merit but does not fully meet PLOS ONE’s publication criteria as it currently stands. Therefore, we invite you to submit a revised version of the manuscript that addresses the points raised during the review process.

Please submit your revised manuscript by Jan 10 2022 11:59PM If you will need more time than this to complete your revisions, please reply to this message or contact the journal office at plosone@plos.org. Please include the following items when submitting your revised manuscript:A rebuttal letter that responds to each point raised by the academic editor and reviewer(s). You should upload this letter as a separate file labeled 'Response to Reviewers'.A marked-up copy of your manuscript that highlights changes made to the original version. You should upload this as a separate file labeled 'Revised Manuscript with Track Changes'.An unmarked version of your revised paper without tracked changes. You should upload this as a separate file labeled 'Manuscript'.If applicable, we recommend that you deposit your laboratory protocols in protocols.io to enhance the reproducibility of your results. Protocols.io assigns your protocol its own identifier (DOI) so that it can be cited independently in the future. For instructions see: https://journals.plos.org/plosone/s/submission-guidelines#loc-laboratory-protocols. Additionally, PLOS ONE offers an option for publishing peer-reviewed Lab Protocol articles, which describe protocols hosted on protocols.io. Read more information on sharing protocols at https://plos.org/protocols?utm_medium=editorial-email&utm_source=authorletters&utm_campaign=protocols.

We look forward to receiving your revised manuscript.

Kind regards,

Academic Editor

PLOS ONE

Important: If there are ethical or legal restrictions to sharing your data publicly, please explain these restrictions in detail. Please see our guidelines for more information on what we consider unacceptable restrictions to publicly sharing data: http://journals.plos.org/plosone/s/data-availability#loc-unacceptable-data-access-restrictions. Note that it is not acceptable for the authors to be the sole named individuals responsible for ensuring data access

Additional Editor Comments (if provided):

Please revise the paper as suggested.

Reviewers' comments:

Reviewer's Responses to Questions

**Comments to the Author**

1. Is the manuscript technically sound, and do the data support the conclusions?

Reviewer #1: Yes

Reviewer #2: Yes

Reviewer #3: Yes

2. Has the statistical analysis been performed appropriately and rigorously? 

Reviewer #1: Yes

Reviewer #2: Yes

Reviewer #3: I Don't Know

3. Have the authors made all data underlying the findings in their manuscript fully available?

Reviewer #1: Yes

Reviewer #2: Yes

Reviewer #3: Yes

4. Is the manuscript presented in an intelligible fashion and written in standard English?

Reviewer #1: Yes

Reviewer #2: Yes

Reviewer #3: Yes

5. Review Comments to the Author

Reviewer #1: Please add key findings and conclusion add in the abstract. Some grammatic error observed in text. read all manuscript and Correct the errors. Some references did not arrange according to format. So, Author should arrange reference

according to PLOS format. Overall, paper reflects good arrangement and is suitable for publication in PLOS Journal

Reviewer #2: 1. Arrange the references in ascending order

2. Cross check the references with the main body text.

3. Minor typographical mistakes needs improvement/correction.

Reviewer #3: The manuscript is well written. Enormous amount of work is done with a sound technical support. The idea of the manuscript is well thought. I find the work can be published with minor revisions. There are some minor issues in writing that should be corrected by a serious revision of the manuscript for English writing. The use of abbreviations should be also be revised. The abbreviations at the start of the sentence should be avoided. The only problem I found in this manuscript is the Discussion part. The discussion is relatively weak. More focus should be given to the principal findings of the data. Therefore, I would suggest particularly a revision of the Discussion part. Rest seems good for publication.

6. PLOS authors have the option to publish the peer review history of their article (what does this mean?). If published, this will include your full peer review and any attached files.

Reviewer #1: No

Reviewer #2: No

Reviewer #3: No

---

## [Author Response · Author response to Decision Letter 0]

4 Jan 2022

Thanks to the reviewers for valuable suggestions. I have incorporated all the suggestions in my manuscript.

---

## [Editor Report · Decision Letter 1]

10 Jan 2022

PONE-D-21-31839R1Residual effect of defeated stripe rust resistance genes/QTLs in bread wheat against prevalent pathotypes of Pucciniastriiformis f. sp. triticiPLOS ONE

Dear Dr. HARPREET SINGH, Thank you for submitting your manuscript to PLOS ONE. After careful consideration, we feel that it has merit but does not fully meet PLOS ONE’s publication criteria as it currently stands. Therefore, we invite you to submit a revised version of the manuscript that addresses the points raised during the review process.

We look forward to receiving your revised manuscript.

Kind regards,

Dr. Muhammad Ishtiaq

Academic Editor

PLOS ONE
---

## [Author Response · Author response to Decision Letter 1]

11 Jan 2022

1. All the figures have been processed using PACE digital diagnostic tool (in previous revision) to ensure that figures meet PLOS requirements.

2. All the references have been updated and corrected according to PLOS requirements.

Changes made in references:

Reference no. 26 was clubbed with Reference no. 25, so it has been moved ahead.

Reference no. 28 was clubbed with Reference no. 27, it has also been moved ahead.

Other minor corrections in the references have been corrected.

References has also been updated in Supplementary table 2 (S2_Table).

---

## [Editor Report · Decision Letter 2]

14 Jan 2022

PONE-D-21-31839R2Residual effect of defeated stripe rust resistance genes/QTLs in bread wheat against prevalent pathotypes of Pucciniastriiformis f. sp. triticiPLOS ONE

Dear Dr. HARPREET SINGH, Ph.D.,

Thank you for submitting your manuscript to PLOS ONE. After careful consideration, we feel that it has merit but does not fully meet PLOS ONE’s publication criteria as it currently stands. Therefore, we invite you to submit a revised version of the manuscript that addresses the points raised during the review process.

We look forward to receiving your revised manuscript.

Kind regards,

Muhammad Ishtiaq

Academic Editor

PLOS ONE

Journal Requirements:

Additional Editor Comments:

The paper requires minor changes, and update and correct it and resend.

---

## [Author Response · Author response to Decision Letter 2]

16 Feb 2022

After careful reading of the paper following changes have been made and accordingly references have been updated.

Additional Editor Comments:

The minor changes and information have been updated.

• Line23: Pst italicized 

• Line47-49 updated: 83 permanently designated stripe rust resistance genes, 71 temporarily designated genes and 363 quantitative trait loci (QTLs); accordingly references have been updated.

Additional requirements

1. All the references have been updated and corrected according to PLOS requirements.

Changes made in references:

• Reference no. 5 and 6 have been removed and replaced with the latest updated references.

Deleted reference no. 5 and 6:

5. McIntosh RA, Dubcovsky J, Rogers WJ, Morris C, Xia XC. Catalogue of gene symbols for wheat: 2017 supplement. [Internet]. 2017. Available from: https://shigen.nig.ac.jp/wheat/komugi/genes/macgene/supplement2017.pdf.

6. Gessese M, Bariana H, Wong D, Hayden M, Bansal U. Molecular mapping of stripe rust resistance gene Yr81 in a common wheat landrace Aus27430. Plant Dis. 2019; 103:1166-1171. doi: 10.1094/pdis-06-18-1055-re

Updated current references: (numbered as 5 and 6)

5. McIntosh RA, Dubcovsky J, Rogers WJ, Xia XC, Raupp WJ. Catalogue of gene symbols for wheat: 2020 supplement. Annu Wheat Newslett. 2020; 66:109-149.

6. Singh K, Batra R, Sharma S, Saripalli G, Gautam T, Singh R, et al. WheatQTLdb: a QTL database for wheat. Mol Genet Genom. 2021; 296(5):1051-1056. doi: 10.1007/s00438-021-01796-9 

• Other minor corrections in the references have been corrected.

---

## [Editor Report · Decision Letter 3]

4 Mar 2022

PONE-D-21-31839R3Residual effect of defeated stripe rust resistance genes/QTLs in bread wheat against prevalent pathotypes of Puccinia striiformis f. sp. triticiPLOS ONE

Dear Dr. HARPREET SINGH,

Thank you for submitting your manuscript to PLOS ONE. After careful consideration, we feel that it has merit but does not fully meet PLOS ONE’s publication criteria as it currently stands. Therefore, we invite you to submit a revised version of the manuscript that addresses the points raised during the review process.

We look forward to receiving your revised manuscript.

Kind regards,

Muhammad Ishtiaq

Academic Editor

PLOS ONE

Journal Requirements:

Additional Editor Comments (if provided):

The reviewers have evaluated your paper and the attached/following corrections are sent herewith it. Correct these and then resubmit for final review and comments.
---

## [Author Response · Author response to Decision Letter 3]

5 Mar 2022

Journal Requirements: 

After reviewing all the references, one reference has been deleted and replaced with latest updated one.

Deleted Reference no. 1

1. Bal RS. Effect of some fungicides and time of fungicidal spray on stripe rust of wheat. J Plant Pest Sci. 2014; 1:39-43.

Updated latest reference no. 1

1. Singh H, Kaur J, Bala R, Singh S, Pannu PPS. Assessment of yield losses due to stripe rust caused by Puccinia striiformis f. sp. tritici on bread wheat cultivars in Punjab. Agric Res J. 2019; 56:698-702. doi: 10.5958/2395-146X.2019.00108.X

All the remaining references are complete and correct.

Additional Editor Comments:

No query from editor was received in this revision. All the queries of editor were addressed in previous revisions.

Reviewer’s comments:

All the queries from reviewer were addressed in the previous revisions.

---

## [Editor Report · Decision Letter 4]

22 Mar 2022

Residual effect of defeated stripe rust resistance genes/QTLs in bread wheat against prevalent pathotypes of Puccinia striiformis f. sp. tritici

PONE-D-21-31839R4

Dear Dr. SINGH,

We’re pleased to inform you that your manuscript has been judged scientifically suitable for publication and will be formally accepted for publication once it meets all outstanding technical requirements.

Kind regards,

Muhammad Ishtiaq

Academic Editor

PLOS ONE

Additional Editor Comments (optional):

Accepted
---

## [Editor Report · Acceptance letter]

24 Mar 2022

PONE-D-21-31839R4 

Residual effect of defeated stripe rust resistance genes/QTLs in bread wheat against prevalent pathotypes of *Puccinia striiformis* f. sp. *tritici*

Dear Dr. Singh:

I'm pleased to inform you that your manuscript has been deemed suitable for publication in PLOS ONE. Congratulations! Your manuscript is now with our production department. 

Kind regards, 

on behalf of

Dr. Muhammad Ishtiaq 

Academic Editor

PLOS ONE